# Mindfulness-Based Interventions and Body Awareness

**DOI:** 10.3390/brainsci12020285

**Published:** 2022-02-18

**Authors:** Marbella Pérez-Peña, Jessica Notermans, Olivier Desmedt, Katleen Van der Gucht, Pierre Philippot

**Affiliations:** 1Human Sciences Sector, Faculty of Psychology and Educational Sciences, School of Psychology, University of Louvain, 1348 Louvain-la-Neuve, Belgium; jessicanotermans@gmail.com (J.N.); olivier.desmedt@uclouvain.be (O.D.); pierre.philippot@uclouvain.be (P.P.); 2Psychological Sciences Research Institute, University of Louvain, 1348 Louvain-la-Neuve, Belgium; 3Leuven Mindfulness Centre, Humanities and Social Sciences Group, Faculty of Psychology and Educational Sciences, KU Leuven, 3000 Leuven, Belgium; katleen.vandergucht@kuleuven.be; 4Fund for Scientific Research, Belgium (FRS-FNRS), 1000 Brussels, Belgium

**Keywords:** mindfulness-based interventions, mindfulness-based cognitive therapy, body awareness, interoception, mechanisms of mindfulness, controlled trial, transdiagnostic, psychological processes

## Abstract

Body awareness (BA) has long been proposed as a working mechanism of mindfulness-based interventions (MBIs), yet research on the mediating role of BA is scarce. Hence, the present study assesses the impact of an 8-week MBI on self-reported and indirect measures of BA, investigates the potential mediating role of BA in the relationship between an MBI and symptomatology, evaluates the impact of an MBI on important psychological processes (i.e., experiential avoidance, rumination, self-efficacy, and self-discrepancy), and explores whether these variables act alongside BA in mediating the relationship between an MBI and symptomatology. A non-randomized controlled trial was conducted with 148 participants (*n* = 89 in the MBI group; *n* = 59 in the control group) who completed questionnaires assessing BA and the above-mentioned psychological processes before and after an MBI. A sub-sample of participants (*n* = 86) completed a task that evaluates BA indirectly. Results showed a significant effect of MBI on the self-reported BA but not on the indirect measure of BA. The MBI significantly reduced symptomatology, and this effect was mediated by regulatory and belief-related dimensions of BA. Multiple mediator models showed a significant mediation via various pathways involving improved BA and various transdiagnostic psychological processes.

## 1. Introduction

Mindfulness-based interventions (MBIs) are programs that employ systematic and sustained training in mindfulness meditation practice as a core methodology [1]. The main therapeutic component of MBIs is the skill of mindfulness, which can be defined as “the awareness that emerges through paying attention on purpose, in the present moment, and nonjudgmentally to the unfolding of experience moment by moment” [2] (p. 145). Ample evidence supports the effectiveness of MBIs in improving a range of physical and psychological conditions as well as quality of life in clinical and non-clinical populations [3,4,5]. One of the theorized working mechanisms by which MBIs exert their positive effects is by improving body awareness [6,7,8].

Although the term “body awareness” (BA) has been defined in different ways throughout the years, it can be broadly defined as paying attention to and being aware of internal bodily sensations [8] (see this reference for a review on the BA construct). More specifically, in the present paper, we use a multidimensional conceptualization of BA: “the subjective, phenomenological aspect of proprioception and interoception that enters conscious awareness, which is modifiable by mental processes including attention, interpretation, appraisal, beliefs, memories, conditioning, attitudes and affect” [9] (p. 1). The BA construct, as defined here, encompasses two important aspects of conscious bodily experience: interoceptive and proprioceptive awareness.

Interoception refers to the sensing, interpreting, and integrating of internal bodily signals by the nervous system at conscious and unconscious levels [10]. Some examples of internal bodily sensations include heartbeat, respiration, satiety, and emotion-related sensations arising from the autonomic nervous system [11]. Khalsa and colleagues [10] outline eight different features of interoception: interoceptive attention (i.e., merely observing body sensations), interoceptive detection (i.e., conscious report of body sensations), interoceptive magnitude (i.e., perceived intensity of body sensations), interoceptive discrimination (i.e., differentiating sensations from different organ systems), interoceptive accuracy (i.e., the ability to precisely monitor one’s sensations), interoceptive sensibility (i.e., one’s perceived tendency to focus on body sensations), interoceptive insight (i.e., confidence about one’s interoceptive accuracy abilities), and self-report scales (i.e., questionnaire assessments of interoception). In light of the definition of BA stated in the previous paragraph, our main interests with regards to interoception are the aspects that are available to conscious awareness and self-reporting, mainly “interoceptive sensibility” and “self-report scales” [10]. In the present paper, we employ the term interoceptive awareness (IAw) to refer to these two components of interoception.

Proprioception is the perception of body position and movements and relies on both physiological (e.g., mechanosensory neurons throughout the body referred to as proprioceptors) and psychological processes (e.g., memory and learning) [12,13]. Like interoception, proprioception is largely unconscious, but some proprioceptive information is accessible to conscious awareness [13]. The conscious perception of body position and movement is known as proprioceptive awareness (PAw) [11] and is the present study’s main interest with regards to proprioception.

An important note must be made regarding the attention styles and regulatory aspects of BA. The literature has identified two attention styles (i.e., the attitude with which one pays attention to bodily sensations): one that is anxiety-driven, evaluative, and avoidant (i.e., maladaptive BA) and another that is mindful, non-judgmental, and accepting (i.e., adaptive BA) [8,14,15]. The first is associated with negative outcomes, such as the generation of aversive emotions [16], hypochondriasis [17], and anxiety disorders [18], whereas the second is associated with positive outcomes, such as increased subjective well-being [19], pain attenuation [20], and reduced severity of post-traumatic stress disorder symptoms [21]. Moreover, several authors have considered how awareness of body sensations can be used to regulate psychological distress [6]. Farb and colleagues [6] propose that sensations (and the affective states associated with them) can be regulated using different strategies: changing the sensation itself, in line with psychological accounts of regulation (e.g., distraction, reappraisal, and suppression) or changing the attitude towards the sensations following contemplative accounts of regulation (e.g., equanimity and acceptance). The present study focuses on BA as characterized by a mindful and accepting attention style and a contemplative regulatory strategy.

By teaching individuals to pay attention to their bodily experiences in an open, accepting, and curious way, MBIs may train adaptive BA [22,23]. BA is an essential element of the exercises taught in MBIs, which involve paying attention to the body at rest (e.g., body scan and breathing meditation) and in movement (e.g., yoga and mindful walking). Existing research on the topic, however, has yielded mixed results. Below, we briefly review the literature on MBIs (and related practices), IAw, and PAw.

The interoception literature has found evidence for the claim that mindfulness practices improve IAw. Several randomized-controlled trials (RCTs) have found a positive impact of mindfulness training on IAw as measured by the Multidimensional Assessment of Interoceptive Awareness (MAIA) [11] in individuals with chronic pain and comorbid depression [24], women in their third trimester of pregnancy [25], and participants with depression [26]. Despite these promising results, two of these studies were underpowered pilot studies [24,25] in need of replication. Furthermore, a non-randomized clinical trial found significant improvements in the IAw (as measured by the MAIA) [11] of healthy individuals after an adapted bodily focused mindfulness training; effects were particularly strong for the regulatory aspects of IAw (i.e., regulating one’s distress by attending to body sensations, regulating attention to and from body sensations, and listening to the body for important information) [27]. Qualitative studies are in line with the aforementioned findings [28,29,30]. These self-report findings are also supported by neuroscientific studies that have found evidence for increased insular activation, the brain region associated with interoception [10,31,32], after an MBI in a general population [33] and in mindfulness meditators [34]. A recent meta-review on the topic confirms these findings by showing that, across various systematic reviews, the insular cortex is the brain area most consistently activated by both interoception and mindfulness [35]. Conversely, studies assessing other facets of interoception have found weak or no associations between mindfulness and interoceptive accuracy and detection [36,37].

The literature assessing proprioception and mindfulness is much less extensive than the literature on mindfulness and interoception; however, existing studies support the claim that mindfulness practices improve proprioception. For example, mindfulness meditation was associated with improved motor performance (i.e., efficient control of motor processes), slower and more accurate body movements, and increased awareness of perceptual-motor conflict in a visuo-motor reaching task with false feedback [38]. This was true for novice meditators who had just completed an MBI and for long-term meditators [38]. Furthermore, an RCT found that elderly women who engaged in an 8-week walking meditation intervention showed improved balance and ankle proprioception (as measured by an ankle reposition test) in comparison to a control group [39]. Lastly, two non-randomized controlled trials found that yoga practices led to improved proprioceptive skills (i.e., higher accuracy of joint position) in healthy individuals [40] and in congenitally blind young people [41].

Taken together, the aforementioned findings suggest that an MBI may improve BA. However, since there are only a few existing studies assessing this relationship, and since many of them are underpowered pilot studies, there is a need to replicate these findings. Furthermore, it is important to explore the mechanisms involved in the relationship between MBIs, BA, and psychological symptomatology in order to understand how MBIs can be effective as well as improved. Very few studies have investigated the mediating mechanisms involved. De Jong and colleagues [24] found that the positive impact of an MBI on depressive symptomatology was mediated by BA (as measured by the MAIA) [11], particularly by the tendency to not distract oneself from unpleasant sensations. However, this study was limited by a small sample size (*N* = 31). Additionally, Fissler and colleagues [26] found that an MBI significantly reduced depressive symptoms, and this was mediated by a serial pathway in which increased BA (as measured by the MAIA) [11] was positively associated with one’s ability to decenter (i.e., stepping out of one’s thoughts and feelings and observing them as temporary and not necessarily related to the self) [42], which was associated with reduced depressive symptoms. However, changes in BA alone did not explain the reduction in symptoms. Considering the limited number of studies that have investigated the mediating role of BA in MBIs’ effect on symptomatology, the field could benefit from a replication of this finding. Moreover, existing research on the topic could be extended by exploring multiple-mediator mechanisms capturing the potential interplay between BA and other psychological variables (i.e., experiential avoidance, rumination, self-efficacy, and self-discrepancy) in explaining the relationship between an MBI and reduced symptomatology. These specific psychological variables were selected because prior reviews and studies have documented their mediating role in explaining the effects of mindfulness-based practices [7,43,44]. Last but not least, considering how challenging it is to measure a multi-faceted construct like BA and taking into account the lack of association between self-report and objective measures of BA found in the literature [45], studies would strongly benefit from using mixed measures of BA (e.g., both self-report and indirect measures of BA) and explore the link between the two.

Taking the above into account, the present study aims to answer the following research questions: (1) What is the impact of an 8-week MBI on self-reported and indirect measures of BA? (2) What is the relationship between scores on self-reported measures of BA and performance on an indirect measure of BA? (3) What is the impact of an MBI on experiential avoidance, rumination, self-efficacy, and self-discrepancy? Finally, (4) what is the impact of an MBI on symptomatology, and is this effect mediated by BA and related psychological processes (i.e., experiential avoidance, rumination, self-discrepancy, and self-efficacy)?

Regarding the first research question, we hypothesized that participants in the MBI group would show significant improvements in BA, as measured by both self-report and indirect measures, in comparison to the control group. More specifically, we hypothesized an increase in the tendency to report body sensations during the recall of positive and negative autobiographical memories, as measured by the indirect BA measure, in all facets of the Functional Body Sensation Questionnaire (FBSQ) [46], and in the MAIA [11] facets of Not-Distracting [24], Attention Regulation [26,27], Emotional Awareness [27], Self-Regulation [24,26,27], Body Listening and Trusting [26,27] in the MBI group in comparison to baseline and to the control group. We did not hypothesize significant changes in the facets of Noticing or Not-Worrying as no prior studies, to our knowledge, have found this effect.

Regarding the second research question, we hypothesized a significant correlation between scores in the indirect measure of BA and the MAIA subscale Emotional Awareness, given that participants talked about emotional memories during the task. Correlations between scores in the indirect measure and other facets of the MAIA and all facets of the FBSQ were computed in an exploratory fashion.

Concerning the third research question, we hypothesized a significant reduction in experiential avoidance (i.e., the tendency to avoid unpleasant internal experiences) [47,48], a significant reduction in unconstructive rumination (i.e., abstract, analytical thinking focused on past and future events) [49,50,51,52,53], a significant increase in constructive rumination (i.e., concrete thinking on one’s present moment experience) [50], a significant increase in general self-efficacy (i.e., confidence in one’s ability to carry out specific behaviors in different domains of life) [54,55], a significant increase in the self-efficacy dimension of emotion regulation (i.e., confidence in one’s ability to regulate one’s emotions) [56,57], a significant reduction in the actual-ideal self-discrepancy gap (i.e., the discrepancy between who people believe they are and who they would like to be) [58] and the distress elicited by this discrepancy [59], no change in the actual-ought self-discrepancy gap (i.e., the discrepancy between who people believe they are and who they believe others would want them to be) [58,59], and a significant reduction in the distress elicited by the actual-ought self-discrepancy in the MBI group in comparison to the control group.

With regards to the fourth research question, we hypothesized that participants in the MBI group would show significant improvements in symptomatology in comparison to the control group and that this effect would be significantly mediated by training-related increases in BA, as measured by the MAIA and FBSQ. Since there is evidence suggesting that BA does not act on its own, but in conjunction with other psychological processes, such as decentering [26], exploratory multiple mediator models were computed with BA as the first mediator and other psychological variables measured in the present study as second mediators (i.e., rumination, experiential avoidance, self-discrepancy, self-efficacy).

The present study’s hypotheses, data analysis plan, and information about the dataset have been pre-registered at the Open Science Framework [60].

## 2. Materials and Methods

### 2.1. Participant Recruitment and Procedure

The study consisted of two groups of French-speaking participants: an intervention group consisting of adults following an 8-week Mindfulness-Based Cognitive Therapy program (MBCT) [23] and a control group consisting of age-, gender-, and education-matched controls. The majority of intervention group participants were recruited on a voluntary basis, following a free information session about MBCT offered by the University of Louvain’s (UCLouvain) Specialized Psychological Consultation Center (CPS). During this information session, participants were informed about the general structure and aims of MBCT as well as the indications and contra-indications for participation. Contra-indications included: suffering from acute depression, severe attention deficit, dissociations, psychotic disorders, substance abuse, or lack of motivation. Participants were also asked to participate in all sessions as well as to practice daily exercises. Motivation and engagement were highlighted as important criteria for participation. A minority of participants was recruited via collaborators who are self-employed practitioners and whose client base is similar to that of the CPS. These participants received the same information as participants recruited via the CPS.

After the information session, interested participants were contacted by their instructors via email. The email included a link giving participants access to a battery of seven questionnaires (through the online Qualtrics platform), which took approximately 40 min to complete. The email mentioned that the study was investigating the effects of MBCT on specific psychological symptoms and mechanisms and specified that questionnaires were to be completed before, or one or two days after, the first MBCT session. In parallel to this email, another email was sent to determine an appropriate time for interested participants to complete a modified version of the Autobiographical Memory Test (mAMT) [61]. Data collected after the questionnaire and task completion comprised the pre-treatment measures (i.e., Time 1). At the end of the 8-week program, participants were asked to fill in the same questionnaires and complete the mAMT a second time, comprising the post-treatment measures (i.e., Time 2). MBCT training was given to various intervention groups over a 2-year time span (between 2016 and 2018).

Control group participants were healthy volunteers recruited via word of mouth, social media, and advertisements at the UCLouvain Psychology faculty. The information distributed to control group participants was similar to the information given to intervention group participants. Control group participants were selected and matched with intervention group participants in terms of age, gender, and level of education. Participants in the control group were also asked to fill in the battery of questionnaires and to complete the mAMT task before and after an 8-week time frame.

This study was conducted in accordance with the Code of Ethics for research involving human participants in the Faculty of Psychology and Educational Sciences of UCLouvain. All participants signed an informed consent form. A debriefing was done at the end of the study (after Time 2 of data collection)—8 to 10 weeks after the start of the training—in order to disclose the full objectives of the study. There was no randomization of the groups and no blinding in the experiment. Inclusion criteria for both groups were: (1) being at least 18 years old, (2) being comfortable speaking and understanding French, (3) having no prior meditation and/or mindfulness experience, and (4) not having any brain injuries and/or cerebral anomalies.

### 2.2. Intervention

The intervention consisted of 2 h weekly group sessions and lasted eight weeks. It was based on the MBCT program described by Segal and colleagues [23], which combines mindfulness practices with elements of cognitive-behavioral therapy. MBCT teaches a non-judgmental, curious, and accepting attitude towards one’s experiences. The present study’s programs were taught by seven MBCT accredited and experienced mindfulness professionals who are also licensed clinical psychologists.

### 2.3. Measures

In order to achieve its aims, the present study employed two self-report questionnaires and one indirect measure of BA. The questionnaires were: the French version of the MAIA [11,62] and the Functional Body Sensation Questionnaire (FBSQ; in the process of being validated) [46]. Both of these self-report measures operationalize BA as defined in the Introduction section, considering mindful and open attention to body sensations as well as how these sensations can be used for self-regulation. They differ in the following ways: (1) the FBSQ focuses on sensations associated with emotions whereas the MAIA evaluates emotion and non-emotion-related sensations; (2) the FBSQ includes a sub-scale, which measures differentiation of emotion-related sensations, which the MAIA does not include; and (3) the MAIA includes sub-scales measuring one’s tendency to not distract oneself from sensations (Not-Distracting), one’s tendency to not react with worry or distress to sensations (Not-Worrying), and one’s experience of the body as safe (Trusting), which the FBSQ does not include. Finally, to address the criticism that using self-reporting to assess BA is limited because participants are asked to report on something they may not be aware of [11], the present study employed an indirect measure of BA (see mAMT below).

#### 2.3.1. Socio-Demographic Variables

Participants were asked to provide information on gender, age, socio-professional status, and level of education. Those in the MBCT group were also asked whether they practiced mindfulness (or other forms of meditation) in the past/present as well as the duration of this practice, whether they took medications, and to provide the name of their MBCT instructor.

#### 2.3.2. The Multidimensional Assessment of Interoceptive Awareness

The MAIA was translated from English to French and validated in a French-speaking sample by Michael and colleagues [62] from the MAIA questionnaire developed by Mehling and colleagues [11]. The MAIA includes 32 items investigating eight dimensions, namely Noticing (i.e., awareness of pleasant, unpleasant, and neutral body sensations), Not-Distracting (i.e., tendency not to ignore or distract oneself from sensations of pain or discomfort), Not-Worrying (i.e., tendency not to worry or feel distress with sensations of pain or discomfort), Attention Regulation (i.e., ability to sustain and control attention to body sensations), Emotional Awareness (i.e., awareness of the connection between body sensations and emotional states), Self-Regulation (i.e., ability to regulate psychological distress by attention to body sensations), Body Listening (i.e., actively listening to the body for insight), and Trusting (i.e., experiencing one’s body as safe and trustworthy). Answers were rated on a six-point Likert scale from 0 (Never) to 5 (Always). The psychometric properties of the MAIA are well-documented [11,63]. Regarding the validity of the MAIA, results suggested that the factorial structure of the MAIA was similar to the English version [62]. Internal consistency in the current sample at baseline was *α* = 0.83 (Noticing), *α* = 0.55 (Not-Distracting), *α* = 0.61 (Not-Worrying), *α* = 0.90 (Attention Regulation), *α* = 0.82 (Emotional Awareness), = 0.80 (Self-Regulation), *α* = 0.87 (Body Listening), and *α* = 0.86 (Trusting).

#### 2.3.3. The Functional Body Sensation Questionnaire

The FBSQ includes 12 items investigating 3 dimensions (each made up of 4 items), namely Perception, Differentiation, and Emotion Regulation. The latter dimension can be understood as using body sensations to regulate emotions. Answers are rated on a five-point Likert scale ranging from 0 (Not at all) to 4 (Completely). A preliminary estimation of the psychometric properties of the questionnaire was done by Pauels and colleagues [46] and suggested good convergent validity. A formal evaluation of its psychometric properties is pre-registered and underway [64]. Internal consistency in the current sample at baseline was *α* = 0.92 (total score), *α* = 0.78 (Perception sub-scale), *α* = 0.86 (Differentiation sub-scale), and *α* = 0.83 (Emotion Regulation sub-scale).

#### 2.3.4. Multidimensional Experiential Avoidance Questionnaire

The Multidimensional Experiential Avoidance Questionnaire (MEAQ) is composed of 62 items rated on a six-point Likert scale ranging from 1 (strongly disagree) to 6 (strongly agree) [65]. It measures avoidance through 6 dimensions, namely Behavioral Avoidance, Procrastination, Distraction/Suppression, Repression/Denial, Distress Aversion (i.e., non-acceptance of or negative attitudes towards distress), and Distress Endurance (i.e., motivation to engage in behaviors consistent with one’s values regardless of distress). The questionnaire was validated in a French-speaking sample [66]. The present study used 38 out of the 62 items because only four dimensions were of interest for this study: Behavioural Avoidance, Procrastination, Distraction/Suppression, and Repression/Denial. Internal consistency in the current sample at baseline was *α* = 0.90 (total score), *α* = 0.88 (Behavioral avoidance), *α* = 0.86 (Procrastination), *α* = 0.86 (Distraction/Suppression), and *α* = 0.85 (Repression/Denial).

#### 2.3.5. Mini Cambridge Exeter Repetitive Thought Scale

The Mini Cambridge Exeter Repetitive Thought Scale (Mini-CERTS) was adapted from the Cambridge Repetitive Thought Scale (CERTS) [67] and translated into French by Douilliez and colleagues [68] and consists of 15 items. Items are rated on a four-point Likert scale ranging from 1 (almost never) to 4 (almost always). The questionnaire evaluates the processing mode of repetitive negative thought based on two dimensions: abstract analytical thinking (i.e., unconstructive rumination) and concrete experiential thinking (i.e., constructive rumination). Assessment of the Mini-CERTS’ psychometric properties in a French-speaking population was performed by Douilliez and colleagues [68]. Internal consistency in the current sample at baseline was *α* = 0.74 (Unconstructive rumination) and *α* = 0.68 (Constructive rumination).

#### 2.3.6. The Self-Efficacy Questionnaire

The Self-Efficacy Questionnaire (S-EQ) was created by Philippot and colleagues [69], following recommendations by Bandura and colleagues [70] in developing self-efficacy scales. The questionnaire comprises 10 items about different life domains such as family, work, relationships, hobbies, etc. Individuals evaluate their level of confidence in their ability to manage each domain on a scale from 0 (not certain at all of my capacity to do this) to 100 (highly certain of my capacity to do this), with 10-point increments. This questionnaire allows clinicians and researchers to identify in which domain self-efficacy is considered the lowest and/or the highest for an individual. The psychometric properties of this newly developed questionnaire have not yet been assessed. Internal consistency in the current sample at baseline was *α* = 0.85.

#### 2.3.7. The Self-Discrepancy Scale

The Self-Discrepancy Scale (S-DS) was developed by Philippot and colleagues [58] and inspired by the Integrated Self-Discrepancy Index [71]. It measures discrepancies between mental representations of the self, namely between the actual self and the ideal or socially-prescribed selves. Participants are asked to estimate the perceived gap between their actual and ideal self and between their actual and socially-prescribed self on a seven-point Likert scale (1 = very close to the ideal to 7 = very far from the ideal), and whether these perceived gaps create distress (1 = no distress to 7 = high distress). The psychometric validation of the scale is described in Philippot and colleagues [58]. Internal consistency in the current sample at baseline was *α* = 0.81.

#### 2.3.8. The Symptom Checklist-90 Revised

The Symptom Checklist-90 Revised (SCL-90-R) was developed by Derogatis and Cleary [72] and later validated and translated in French by Pariente and colleagues [73]. It is composed of 90 items measuring psychopathological symptoms in terms of nine dimensions. Items are rated on a five-point Likert scale ranging from 0 (Not at all) to 4 (Extremely). The present study only calculated the Global Severity Index (GSI) to measure general symptom severity. Internal consistency in the current sample at baseline was *α* = 0.96.

#### 2.3.9. Modified Autobiographical Memory Test

A sub-sample of participants (*n* = 86; *n* = 43 in the control group; *n* = 43 in the MBCT group) completed the mAMT, which measures the spontaneous tendency to report body sensations when reporting emotional personal events. The mAMT can be most precisely described as a memory task that can be used to assess the attention allocated to body sensations by counting the number of spontaneous references to body sensations in the memory report. The task is best categorized as an indirect self-reporting measure because, during the task, participants are not explicitly asked to report body sensations, but report of body sensations is an outcome of interest. In the original version of the mAMT [61] and its validated French version [74], participants are given 10 cue words (five negative and five positive) and are asked to retrieve and describe a specific personal memory (i.e., the memory of an event personally experienced that lasted less than 24 h) [75] in response to each word. Next, supplementary information is asked about the emotional intensity attributed to the event (measured on a scale of 0 to 10) and when the event happened. The researcher then codes the memories as specific (i.e., concerning an event that is precisely situated and lasting less than 24 h), extended (i.e., concerning an event lasting longer than 24 h such as “my university years”), generic (i.e., concerning repetitive events such as weekly dance lessons), or omissions (i.e., memories not lived personally and not about the past) [76]. To calculate scores, specific memories receive three points, extended memories receive two points, generic memories receive one point, and omissions receive zero points. Total points are then summed per participant per valence (negative or positive memories).

The mAMT introduced in the present study includes an extra step called the Autobiographical Memory Description Task, which is administered after the AMT. In this part of the task, participants are reminded of six out of the ten keywords for which they retrieved a memory, and are subsequently invited to give as many details as possible about their state during the event constituting the memory. The researcher then codes the participant’s description for mention of body sensations (i.e., details about physical states and non-verbal expression), internal states (i.e., cognitive or emotional details), sensory details (i.e., details linked to the 5 senses), and contextual details (i.e., external, physical or spatiotemporal conditions, people/things that were there, or historical context of the event). Scores are then obtained by summing the mention of details per category. The score for the mention of body sensations can be interpreted as the number of spontaneous references to body sensations and can be used as a proxy for the attention allocated to the body while describing an emotional experience.

### 2.4. Data Analysis

Differences in participants’ characteristics at baseline were assessed with independent sample *t*-tests for continuous variables and Chi-square tests for categorical variables.

Multilevel linear models were used to assess intervention effects because the data is nested [77]. Participants are nested within mindfulness instructors (*n* = 7). Hence, the independence of errors assumption is violated, which rules out the use of statistical tests in which this assumption must be true (i.e., ANOVA, regression, *t*-tests) [77]. Multilevel analyses can model the variation resulting from contextual variables (i.e., different mindfulness instructors), allowing us to overcome the lack of independence [77]. Since our primary interest was an association between Level-1 variables in data with 2-level hierarchy (i.e., patients nested within mindfulness instructors), centering within clusters was used, as suggested by Enders and Tofighi [78].

The R package *nlme* [79] was used to compute multilevel models. The steps outlined in Field and Wright (2011) [77] were followed to build the models. Different models were computed (i.e., first with random effects on the intercept, and then with random effects on both intercept and slope), and model fit was assessed with the Akaike’s Information Criterion (AIC). The model with the lowest AIC was selected for reporting.
(1)$$Model\;1\;\left(random\;intercept\right):\;Y_{ij}=\beta_{0j}+\beta_{1}Intervention_{ij}+\beta_{2}Base\_Y_{ij}+r_{ij}$$
(2)$$Model\;2\;\left(random\;intercept\;and\;slope\right):\;Y_{ij}=\beta_{0j}+\beta_{1j}Intervention_{ij}+\beta_{2}Base\_Y_{ij}+r_{ij}$$

The above equations represent a 2-level model in which Level-1 was represented by the participants and Level-2 by the mindfulness instructors. The only difference between model 1 and model 2 was that model 2 had random effects on the slope, whereas model 1 did not. $Y_{ij}$ denotes the outcome of the *j*-th mindfulness instructor for the *i*-th participant. Separate models were generated with the following outcome measures: post-intervention BA as measured by the FBSQ (total score and facets), MAIA (facets), and the mAMT; post-intervention experiential avoidance (total score and facets); post-intervention actual-ideal self-discrepancy gap; post-intervention actual-ideal self-discrepancy distress; post-intervention actual-ought self-discrepancy gap; post-intervention actual-ought self-discrepancy distress; post-intervention unconstructive rumination; post-intervention constructive rumination; post-intervention self-efficacy, and post-intervention symptomatology. $Base\_Y_{ij}$ are the baseline levels of each of the aforementioned outcome variables. The residual is represented by $r_{ij}$. The treatment group is denoted by the categorical predictor $Intervention_{ij}$ with 0 being the control condition and 1 being the MBCT condition.

Correlations among scores in the FBSQ (total score and facets), MAIA (facets), and the mAMT were computed separately for both time points in order to answer research question 2.

Finally, pre- and post-change scores were calculated for relevant variables and included in a simple mediator model. To assess the mediating role of changes in BA in the effect of MBCT on symptomatology, we used the steps outlined in Yzerbytz and colleagues [80]. According to them, the method that achieves the best balance between power and Type I error is the component method by which one assesses the significance of the component paths of the indirect effects individually first, and then together. The significance of the individual component paths was assessed using joint significance testing as the data was normally distributed. The significance of the indirect effect was then assessed by Monte Carlo confidence intervals. The R package, JSmediation developed by Batailler and colleagues [81], was used to compute these simple mediation models.

Lastly, exploratory multiple-mediator models were calculated using the R package Lavaan. The dimensions of BA that were significantly affected by the MBI were included in separate serial mediator models with BA as a first mediator and experiential avoidance, unconstructive rumination, emotion regulation self-efficacy, actual-ideal self-discrepancy gap, actual-ideal self-discrepancy gap distress, and actual-ought self-discrepancy gap distress as second mediators.

## 3. Results

### 3.1. Participant Characteristics

A total of *n* = 148 (*n* = 59 in the control group; *n* = 89 in the MBCT group) participants were included in the analyses. A sub-sample of participants *n* = 86 (*n* = 43 in the control group; *n* = 43 in the MBCT group) also completed the mAMT. This sub-sample was a convenience sample of participants who were willing and able to complete the task within the specific time frame following recruitment. For details on the flow of participants throughout the study, please see Figure 1. For more details on how the data was pre-processed, please see the pre-registration section: “Prior work based on this dataset” [60].

Participant characteristics are shown in Table 1. Most participants in the experimental group suffered from anxiety and mood disorders. There were no significant differences between the MBCT and control groups in most demographic variables, with the exception of employment status. There were several significant differences in average baseline scores between the two groups (see Table 2).

For the sub-sample who completed the mAMT (*n* = 86), across groups, participants were on average 44 years old (*M* = 44.09, *SD* = 12.99), and college graduates (93% completed higher education studies). Most participants were female (61.6%) and employed (65.13%). In the subsample, 20.94% of participants were taking psychotropic medications. This subsample did not differ from the sample who did not complete the mAMT in terms of age, gender, or medication use. However, there were significant baseline differences in employment status, *X*^2^ (2) = 13.76, *p* = 0.001, with the mAMT group containing fewer students and more people in the employed and other categories. There was also a significant difference in the level of education, *X*^2^ (2) = 12.49, *p* = 0.002, with the mAMT group containing more participants who completed university studies. Concerning baseline levels of the outcome and process variables measured in the present study, there were no significant differences between the two groups except in the case of the FBSQ total and dimension scores (the mAMT group scored significantly higher) and the Distraction/Suppression facet of the MEAQ (the mAMT group scored significantly lower).

There was 3% missing data, with 14 out of the 148 participants who did not complete all questionnaires at a given time point. According to Bennett [82], less than 10% of missing data is not problematic and is not likely to bias the statistical analyses. All available data were used for the analyses.

### 3.2. Outliers

A robust Mahalanobis distance with shrinkage estimators was used to detect multivariate outliers [83]. Eleven outliers were detected for the questionnaire data, and three outliers were detected for the mAMT data. Analyses were run with and without outliers. All results were the same except for: the intervention effect on the mAMT’s specific positive memories, which was significant with outliers but not significant without outliers; the simple mediation model with the MAIA facet Trusting as mediator, which was significant with outliers and not significant without outliers; the multiple mediator model with MAIA Attention Regulation and Emotion Regulation Self-Efficacy as mediators, which was significant with outliers and not significant without outliers; and the multiple mediator model with MAIA Self-Regulation and Actual-Ought Self-Discrepancy Gap Distress as mediators, which was not significant with outliers and significant without outliers. Results with outliers are reported below. In the case of results that changed in the sensitivity analysis, the results of analyses with and without outliers are reported.

### 3.3. Multilevel Analyses: Intervention Effect Models

The multilevel models reported here are the random-intercept models because they had the lowest AIC. Multilevel model results can be found in Table 3.

#### 3.3.1. What Is the Impact of an MBI on Self-Report and Indirect Measures of BA?

The fixed effects of the random-intercept models (see Table 3) indicated that the intervention had a significant effect on post-treatment average FBSQ scores and all facets of the FBSQ (i.e., Perception, Differentiation, and Regulation). There was also a significant effect of intervention on the MAIA facets of Noticing, Not-Worrying, Attention Regulation, Emotional Awareness, Self-Regulation, Body Listening, and Trusting. No significant effect of intervention was found on post-treatment MAIA Not-Distracting.

Concerning the adapted mAMT, multilevel models revealed that the intervention had a significant effect on post-treatment-specific positive memories but no significant effect on post-treatment-specific negative memories. There was also no significant effect of intervention on the post-treatment mention of body sensations, internal states, sensory details, or contextual details during recall of positive or negative memories.

Significant effects of the intervention were above and beyond the effect of pre-intervention levels.

#### 3.3.2. What Is the Impact of an MBI on Symptomatology?

Multilevel models indicated that the intervention had a significant effect on post-treatment symptomatology after controlling for pre-intervention levels (see Table 3).

#### 3.3.3. What Is the Impact of an MBI on Experiential Avoidance, Self-Discrepancy, Rumination, and Self-Efficacy?

Random-intercept models (see Table 3) indicated that intervention significantly reduced post-treatment total Experiential Avoidance, Behavioral Avoidance, Procrastination, and Distraction/Suppression as measured by the MEAQ. However, no significant effect of intervention was found on the MEAQ facet of Repression/Denial. The intervention also significantly reduced post-treatment actual-ideal self-discrepancy gap, actual-ideal self-discrepancy gap distress, and actual-ought self-discrepancy gap distress. No significant effect of intervention was found on the post-treatment actual-ought self-discrepancy gap. Furthermore, intervention significantly reduced post-treatment unconstructive rumination, and there was a marginally significant trend towards an increase in post-treatment constructive rumination. Lastly, the models showed that intervention significantly increased post-treatment general self-efficacy, and emotion regulation self-efficacy. All significant effects were above and beyond the effects of pre-intervention levels.

### 3.4. Correlations between Self-Report and Indirect Measures

There were weak correlations between the FBSQ and MAIA scores and the scores in the mAMT at the pre-treatment phase across groups (*r* ≤ 0.22). At post-treatment, most correlations between the FBSQ and the mAMT task were weak as well (*r* ≤ 0.26) except in the case of the mention of body sensations during recall of positive memories and FBSQ differentiation (*r* = 0.31, *p* = 0.007). In the case of correlations between the MAIA and the mAMT, most correlations were small (*r* ≤ 0.27) except for: Attention Regulation and mention of body sensations during recall of positive memories (*r* = 0.37, *p* = 0.001), Attention Regulation and mention of body sensations during recall of negative memories (*r* = 0.30, *p* = 0.009), Emotional Awareness and mention of body sensations during recall of positive memories (*r* = 0.33, *p* = 0.003), and Self-Regulation and mention of body sensations during recall of positive memories (*r* = 0.34, *p* = 0.002).

### 3.5. Mediation Analyses

#### 3.5.1. Is the Impact of an MBI on Symptomatology Mediated by BA?

Separate simple mediation models were tested with all BA dimensions that were significantly impacted by the MBI (i.e., all dimensions of the FBSQ and all MAIA facets except for Not-Distracting). Only the significant models are reported here. All models had intervention as the independent variable, change in a particular facet of BA as a mediator, and change in symptomatology as the dependent variable. Post-minus pre-treatment change scores were used in the mediation models. All models reported below showed a significant effect of intervention (MBCT or control) on symptomatology, *c* = −0.394, *t*(134) = −6.00, *p* < 0.001.

The first significant model included MAIA Attention Regulation as mediator. Joint significance testing [80] revealed a significant effect of intervention on Attention Regulation, *a* = 1.354, *t*(143) = 8.67, *p* < 0.001, and a significant effect of Attention Regulation on symptomatology controlling for intervention, *b* = −0.105, *t*(133) = 2.98, *p* = 0.003. The effect of intervention on symptomatology after controlling for Attention Regulation was still significant but to a lesser degree, *c*’ = −0.252, *t*(133) = 3.15, *p* = 0.002. Consistent with this result, the Monte Carlo confidence interval for the indirect effect did not contain 0, CI95% (−0.246; −0.0443) indicating that Attention Regulation partially mediated the effect of the intervention on symptomatology.

The second significant model included MAIA Self-Regulation as mediator. Joint significance testing [80] revealed a significant effect of intervention on Self-Regulation, *a* = 1.842, *t*(143) = 11.24, *p* < 0.001, and a significant effect of Self-Regulation on symptomatology controlling for intervention, *b* = −0.086, *t*(133) = 2.47, *p* = 0.015. The effect of intervention on symptomatology after controlling for Self-Regulation was still significant but to a lesser degree, *c*’ = −0.227, *t*(133) = 2.42, *p* = 0.017. Consistent with this result, the Monte Carlo confidence interval for the indirect effect did not contain 0, CI95% (−0.285; −0.0325) indicating that Self-Regulation partially mediated the effect of the intervention on symptomatology.

The third significant model included MAIA Body Listening as mediator. Joint significance testing [80] revealed a significant effect of intervention on Body Listening, *a* = 1.392, *t*(143) = 7.94, *p* < 0.001, and a significant effect of Body Listening on symptomatology controlling for intervention, *b* = −0.079, *t*(133) = 2.33, *p* = 0.021. The effect of intervention on symptomatology after controlling for Listening was still significant, *c*’ = −0.283, *t*(133) = 3.51, *p* < 0.001. The Monte Carlo confidence interval for the indirect effect did not contain 0, CI95% (−0.214; −0.0198) indicating that Body Listening partially mediated the effect of the intervention on symptomatology.

The final significant model included MAIA Trusting as mediator. Joint significance testing [80] revealed a significant effect of intervention on Trusting, *a* = 0.955, *t*(143) = 5.37, *p* < 0.001, and a significant effect of Trusting on symptomatology controlling for intervention, *b* = −0.061, *t*(133) = 2.00, *p* = 0.047. The effect of intervention on symptomatology after controlling for Trusting was still significant, *c*’ = −0.332, *t*(133) = 4.61, *p* < 0.001. The Monte Carlo confidence interval for the indirect effect did not contain 0, CI95% (−0.128; −0.00174) indicating that Trusting partially mediated the effect of the intervention on symptomatology. However, this result was not robust in the sensitivity analysis. Analyses without outliers revealed a significant effect of intervention on Trusting, *a* = 0.931, *t*(133) = 5.19, *p* < 0.001, and a non-significant effect of Trusting on symptomatology controlling for intervention, *b* = −0.052, *t*(123) = 1.72, *p* = 0.087. The effect of intervention on symptomatology after controlling for Trusting was still significant, *c*’ = −0.405, *t*(123) = 5.85, *p* < 0.001. The Monte Carlo confidence interval for the indirect effect contained 0, CI95% (−0.11; 0.008) indicating that Trusting did not mediate the effect of the intervention on symptomatology.

#### 3.5.2. Is the Impact of an MBI on Symptomatology Mediated by BA Alongside Other Psychological Variables?

In an exploratory fashion, separate multiple mediator models were calculated with intervention (MBCT vs. control) as the independent variable, change in symptomatology as the dependent variable, and change in aspects of BA that were significant in the simple mediator models (i.e., Attention Regulation, Self-Regulation, Body Listening, and Trusting) as the first mediator, and a transdiagnostic psychological process as a second mediator. A total of six sets of models were calculated in this way. For an overview and summary of the models, please refer to Figure 2. As with the above models, there was a significant effect of intervention (MBCT or control) on symptomatology, *c* = −0.394, *t*(134) = −6.00, *p* < 0.001. Only significant models are reported below.

The first set of models was tested with intervention (MBCT vs. control) as the independent variable, change in symptomatology as the dependent variable, change in aspects of BA that were significant in the simple mediator models (i.e., Attention Regulation, Self-Regulation, Body Listening, and Trusting) as the first mediator, and experiential avoidance total score as the second mediator.

Results revealed a significant indirect effect of intervention on symptomatology through Attention Regulation, *a1 × b1* = −0.132, *z* = −2.75, *p* = 0.006, CI95% (−0.225; −0.038), and Experiential Avoidance *a2 × b2* = −0.103, *z* = −3.00, *p* = 0.003, CI95% (−0.17; −0.036), with a total indirect effect of (*a1 × b1*) + (*a2 × b2*) = −0.235, *z* = −4.07, *p* < 0.001, CI95% (−0.348; −0.122).

There was also a significant indirect effect of intervention on symptomatology through Self-Regulation, *a1 × b1* = −0.156, *z* = −2.39, *p* = 0.017, CI95% (−0.284; −0.028), and Experiential Avoidance, *a2 × b2* = −0.105, *z* = −3.04, *p* = 0.002, CI95% (−0.173; −0.037), with a total indirect effect of (*a1 × b1*) + (*a2 × b2*) = −0.261, *z* = −3.58, *p* < 0.001, CI95% (−0.405; −0.118).

A second set of models included intervention (MBCT vs. control) as the independent variable, change in symptomatology as the dependent variable, and change in aspects of BA that were significant in the simple mediator models (i.e., Attention Regulation, Self-Regulation, Body Listening, and Trusting) as the first mediator, and unconstructive rumination as a second mediator.

Results revealed a significant indirect effect of intervention on symptomatology through Attention Regulation, *a1 × b1* = −0.115, *z* = −2.49, *p* = 0.013, CI95% (−0.206; −0.024), and Unconstructive Rumination, *a2 × b2* = −0.145, *z* = −3.70, *p* < 0.001, CI95% (−0.222; −0.068), with a total indirect effect of (*a1 × b1*) + (*a2 × b2*) = −0.261, *z* = −4.47, *p* < 0.001, CI95% (−0.375; −0.146).

There was also a significant indirect effect of intervention on symptomatology through Self-Regulation, *a1 × b1* = −0.142, *z* = −2.23, *p* = 0.026, CI95% (−0.266; −0.017), and Unconstructive Rumination, *a2 × b2* = −0.15, *z* = −3.77, *p* < 0.001, CI95% (−0.228; −0.072), with a total indirect effect of (*a1 × b1*) + (*a2 × b2*) = −0.291, *z* = −3.99, *p* < 0.001, CI95% (−0.434; −0.149).

A third set of models was tested with intervention (MBCT vs. control) as the independent variable, change in symptomatology as the dependent variable, change in aspects of BA that were significant in the simple mediator models (i.e., Attention Regulation, Self−Regulation, Body Listening, and Trusting) as the first mediator, and Emotion Regulation Self-Efficacy as the second mediator.

Results revealed a significant indirect effect of intervention on symptomatology through Attention Regulation, *a1 × b1* = −0.105, *z* = −2.27, *p* = 0.023, CI95% (−0.196; −0.014), and Emotion Regulation Self−Efficacy *a2 × b2* = −0.187, *z* = −3.96, *p* < 0.001, CI95% (−0.279; −0.094), with a total indirect effect of (*a1 × b1*) + (*a2 × b2*) = −0.291, *z* = −4.74, *p* < 0.001, CI95% (−0.412; −0.171). However, this result was not robust as the sensitivity analysis without outliers revealed a non-significant indirect effect of intervention on symptomatology through Attention Regulation, *a1 × b1* = −.075, *z* = −1.67, *p* = 0.094, CI95% (−0.162; −0.013), and a significant effect through Emotion Regulation Self-Efficacy *a2 × b2* = −0.175, *z* = −3.86, *p* < 0.001, CI95% (−0.264; −0.086), with a total indirect effect of (*a1 × b1*) + (*a2 × b2*) = −0.250, *z* = −4.27, *p* < 0.001, CI95% (−0.364; −0.135).

A fourth set of models was tested with intervention (MBCT vs. control) as the independent variable, change in symptomatology as the dependent variable, change in aspects of BA that were significant in the simple mediator models (i.e., Attention Regulation, Self-Regulation, Body Listening, and Trusting) as the first mediator, and the actual-ideal self-discrepancy gap as the second mediator.

There was a significant indirect effect of intervention on symptomatology through Attention Regulation, *a1 × b1* = −0.124, *z* = −2.52, *p* = 0.012, CI95% (−0.220; −0.027), and Actual-Ideal Self-Discrepancy Gap *a2 × b2* = −0.066, *z* = −2.18, *p* = 0.029, CI95% (−0.125; −0.007), with a total indirect effect of (*a1 × b1*) + (*a2 × b2*) = −.394, *z* = −6.04, *p* < 0.001, CI95% (−0.296; −0.083).

There was a significant indirect effect of intervention on symptomatology through Body Listening, *a1 × b1* = −0.103, *z* = −2.15, *p* = 0.031, CI95% (−0.196; −0.009), and Actual-Ideal Self-Discrepancy Gap *a2 × b2* = −0.073, *z* = −2.39, *p* = 0.017, CI95% (−0.133; −0.013), with a total indirect effect of (*a1 × b1*) + (*a2 × b2*) = −0.176, *z* = −3.18, *p* < 0.001, CI95% (−0.285; −0.068).

A fifth set of models was tested with intervention (MBCT vs. control) as the independent variable, change in symptomatology as the dependent variable, change in aspects of BA that were significant in the simple mediator models (i.e., Attention Regulation, Self-Regulation, Body Listening, and Trusting) as the first mediator, and actual-ideal self-discrepancy gap distress as the second mediator.

There was a significant indirect effect of intervention on symptomatology through Attention Regulation, *a1 × b1* = −0.122, *z* = −2.80, *p* = 0.005, CI95% (−0.208; −0.036), and Actual-Ideal Self-Discrepancy Gap Distress *a2× b2* = −0.109, *z* = −3.08, *p* = 0.002, CI95% (−0.178; −0.040), with a total indirect effect of (*a1 × b1*) + (*a2 × b2*) = −0.231, *z* = −4.14, *p* < 0.001, CI95% (−0.340; −0.121).

There was also a significant indirect effect of intervention on symptomatology through Self-Regulation, *a1 × b1* = −0.135, *z* = −2.26, *p* = 0.024, CI95% (−0.252; −0.018), and Actual-Ideal Self-Discrepancy Gap Distress, *a2 × b2* = −0.109, *z* = −3.08, *p* = 0.002, CI95% (−0.179; −0.040), with a total indirect effect of (*a1 × b1*) + (*a2 × b2*) = −0.244, *z* = −3.55, *p* < 0.001, CI95% (−0.379; −0.109).

There was a significant indirect effect of intervention on symptomatology through Body Listening, *a1 × b1* = −0.098, *z* = −2.28, *p* = 0.023, CI95% [−0.181; −0.014], and Actual-Ideal Self-Discrepancy Gap Distress *a2 × b2* = −0.110, *z* = −3.09, *p* = 0.002, CI95% [−0.180; −0.040], with a total indirect effect of (*a1 × b1*) + (*a2 × b2*) = −0.208, *z* = −3.75, *p* < 0.001, CI95% [−0.317; −0.099].

A sixth set of models was tested with intervention (MBCT vs. control) as the independent variable, change in symptomatology as the dependent variable, change in aspects of BA that were significant in the simple mediator models (i.e., Attention Regulation, Self-Regulation, Body Listening, and Trusting) as the first mediator, and actual-ought self-discrepancy gap distress as the second mediator.

There was a significant indirect effect of intervention on symptomatology through Attention Regulation, *a1 × b1* = −0.137, *z* = −2.97, *p* = 0.003, CI95% (−0.227; −0.046), and Actual-Ought Self-Discrepancy Gap Distress *a2 × b2* = −0.083, *z* = −2.74, *p* = 0.006, CI95% (−0.142; −0.024), with a total indirect effect of (*a1 × b1*) + (*a2 × b2*) = −0.219, *z* = −3.99, *p* < 0.001, CI95% (−0.327; −0.112).

There was a non-significant indirect effect of intervention on symptomatology through Self-Regulation, *a1 × b1* = −0.121, *z* = −1.905, *p* = 0.057, CI95% (−0.246; 0.003), and a significant effect through the Actual-Ought Self-Discrepancy Gap Distress *a2 × b2* = −0.079, *z* = −2.68, *p* = 0.007, CI95% (−0.138; −0.021), with a total indirect effect of (*a1 × b1*) + (*a2 × b2*) = −0.201, *z* = −2.93, *p* = 0.003, CI95% (−0.335; −0.066). However, the sensitivity analysis without outliers showed a significant indirect effect of intervention on symptomatology through Self-Regulation, *a1 × b1* = −0.132, *z* = −2.20, *p* = 0.03, CI95% (−0.250; −0.014), and Actual-Ought Self-Discrepancy Gap Distress *a2× b2* = −0.08, *z* = −2.77, *p* = 0.006, CI95% (−0.136; −0.023), with a total indirect effect of (*a1 × b1*) + (*a2 × b2*) = −0.212, *z* = −3.29, *p* = 0.001, CI95% (−0.338; −0.086).

## 4. Discussion

The present study investigated the impact of MBCT on BA as measured by self-report questionnaires and an indirect BA task in a heterogeneous clinical adult sample. The association between self-report and indirect measures of BA was evaluated. Furthermore, the potential role of various facets of BA in explaining the reduction in symptomatology observed after MBCT was assessed. Last but not least, the present study evaluated the impact of MBCT on transdiagnostic psychological processes (i.e., experiential avoidance, rumination, self-efficacy, and self-discrepancy) and explored the mediating role of BA alongside these psychological processes in explaining the reduction in symptomatology after MBCT.

In accordance with the initial hypotheses, there was a significant increase in most dimensions of self-reported BA and a significant reduction in psychological symptomatology after the intervention in the MBCT group in comparison to the control group. Moreover, as predicted, the reduction in symptomatology was partially mediated by changes in four dimensions of self-reported BA, mainly Attention Regulation, Self-Regulation, Body Listening, and Trusting. Lastly, as hypothesized, there was a significant improvement in most of the transdiagnostic processes in the MBCT group in comparison to the control group (i.e., experiential avoidance and three of its dimensions, unconstructive rumination, general self-efficacy, emotion regulation self-efficacy, the actual-ideal self-discrepancy gap and the distress caused by this gap, and actual-ought self-discrepancy gap distress). Exploratory analyses further revealed that BA acts alongside several of these transdiagnostic variables in explaining the positive effect of MBCT on psychological symptoms. Contrary to our hypotheses, there was no effect of the intervention on the mAMT, an indirect measure of BA, and most correlations between mAMT scores and BA questionnaire scores were weak. The results are discussed for each research question separately below.

### 4.1. What Is the Impact of an MBI on Self-Report and Indirect Measures of BA?

The present study revealed a significant increase in all dimensions of the FBSQ and on all dimensions of the MAIA (with the exception of Not-Distracting) in the MBCT group in comparison to the control group. Compared to control group participants, those who completed the MBCT reported an increased ability to: notice and perceive comfortable, uncomfortable, and neutral body sensations (FBSQ Perception and MAIA Noticing); regulate attention to body sensations (MAIA Attention Regulation); link body sensations and emotions in a differentiated way (MAIA Emotional Awareness and FBSQ Differentiation); use awareness of body sensations to regulate psychological distress (MAIA Self-Regulation and FBSQ Regulation); not worry when faced with sensations of pain or discomfort (MAIA Not-Worrying); listen to the body for insights (MAIA Body Listening); and experience the body as trustworthy (MAIA Trusting). These findings are in line with prior studies showing significant improvements in various MAIA facets after an MBI in clinical [24,26,84] and non-clinical populations [27,85].

Improvements in the facet of Noticing were surprising because prior studies failed to find this effect [24,26,27,85]. The absence of an effect on Noticing has been previously explained by the absence of a significant effect of MBIs on interoceptive accuracy [86]. However, the fact that subjective and objective measures of BA often do not correlate [45] sheds doubt on this explanation of an absence of an effect on Noticing. The present study shows that participants in a heterogeneous clinical sample report being able to notice body sensations of different valences after mindfulness training, which is in line with the skills cultivated in MBIs. Mindfulness meditation exercises (particularly the body scan) continuously invite participants to notice the presence or absence of body sensations [23], which may train the skill of Noticing. There are several potential reasons for why some studies did not find an effect on Noticing. First, some studies lacked adequate power [24] (*N* = 29), which may have prevented them from finding an effect. Second, some studies were conducted on a population of healthy adults [85], meaning there may have been little room for improvement. Baseline levels of Noticing in the mindfulness group in the study by D’Antoni and colleagues [85] were higher (3.19) than baseline levels in the present study (2.44). Finally, other studies used shorter or adapted versions of an MBI [26,27], which may have lacked the assiduous training employed in standard 8-week courses, thereby diminishing the effect of the intervention on Noticing.

Improvements in Not-Worrying contradict prior studies that found no effect on this facet of BA [26,27,84]. However, our results are in line with a more recent study that found a significant effect of a 7-week Mindfulness-Oriented Meditation intervention on Not-Worrying in a sample of healthy individuals [85]. There are several potential explanations for the inconsistency of results. As mentioned above, some studies use shorter or adapted versions of an MBI [26,27], which may not target the ability to skillfully relate to difficult and painful experiences. The present study used the complete, standardized MBCT program, which includes sessions and meditations on dealing with difficult experiences, and may help explain the improvements in Not-Worrying observed in the present study. Another reason why there may be inconsistent results is that the MAIA sub-scale of Not-Worrying has been shown to have poor internal consistency in various studies [11,26,27], including the present study (*α* = 0.61).

Contrary to our hypotheses, there was no significant effect of MBCT on the tendency to ignore or distract oneself from sensations of discomfort or pain (MAIA Not-Distracting) as compared to the control group. This contradicts De Jong and colleagues’ [24] findings that Not-Distracting not only improves after MBCT but also mediates the effect of MBCT on depressive symptomatology. One would expect that MBCT would significantly impact Not-Distracting as one of the aims of the training is to teach people to approach their internal experience in a curious and gentle way [23], which is at odds with the strategy of distraction. Furthermore, the present study shows that MBCT significantly reduced participants’ experiential avoidance and, more specifically, their tendency to avoid experience via distraction and suppression, which is not congruent with the absence of an effect on Not-Distracting. One of the potential reasons for this null finding is that the Not-Distracting sub-scale exhibited poor internal consistency in prior studies [11] and in the present study as well (*α* = 0.55). Therefore, this result must be interpreted with caution.

Contrary to our hypotheses, there was no significant impact of MBCT on BA as measured by the mAMT. More specifically, there was no effect on the mention of body sensations during autobiographical memory recall. This is in line with prior research using other measurement types, such as behavioral measures, which found no improvements in interoceptive accuracy [10] in meditators [36,87,88], completers of an MBI [86], nor individuals with high trait mindfulness [86]. With regards to the mAMT used in the present study, the lack of an effect may be due to numerous reasons. First, since the adapted version of the task has not yet been validated, the exact construct it is measuring is still unclear. There are many different processes potentially involved in the tendency to spontaneously report one’s bodily state while reporting an emotional autobiographical memory. Some of these underlying processes can be interoceptive, such as an improved ability to attend to and detect body sensations. However, other non-interoceptive processes should be considered, such as participants’ autobiographical memory encoding and recall capacities, as well as their ability (e.g., vocabulary) and willingness to communicate about their body sensations with a stranger. Hence, there is a multiplicity of concurrent variables that may be blurring the effect. Second, the task may lack sensitivity because participants were only asked to discuss three positive and three negative memories. More instances may be needed per category of memories in order to increase the task’s sensitivity. Last but not least, the task’s sensitivity may be reduced because participants are asked to be as specific as possible, which may have interfered with how they would naturally behave. Perhaps if there was less of an emphasis on specificity, the task would be more sensitive to participants’ spontaneous behavior and hence more likely to pick up differences between the MBCT group and control group.

### 4.2. How Do Results in the Indirect Measure of BA Correlate with Results in Self-Report Measures of BA?

Results only partially supported the hypothesis that performance on the mAMT would be correlated with scores on BA questionnaires. We had specifically hypothesized a significant correlation between performance on the task and the MAIA subscale Emotional Awareness, given that participants talked about body sensations during emotional memories. There was some evidence for this hypothesis as there was a medium correlation between mention of body sensations during recall of positive memories and Emotional Awareness, suggesting that the more aware participants were of the connection between body sensations and emotions, the more they reported body sensations linked to a positive memory. Furthermore, the more participants were able to regulate their attention to body sensations, differentiate the states suggested by these sensations, and use these sensations to regulate themselves, the more they reported body sensations when talking about positive memories. This suggests that increased adaptive BA could lead to more awareness and, therefore, reporting of positive body sensations. This can be explained by the fact that an increased ability to regulate one’s attention towards body sensations may lead to more awareness of positive body sensations as people can consciously and deliberately place their attention on positive sensations. Lastly, these correlations point to a congruence between the mAMT and specific facets of self-reported BA.

Globally, however, all other correlations between self-reported BA and performance on the task were small and not significant, which may be due to the task’s lack of sensitivity or to the fact that the task and questionnaires are measuring different constructs. These findings are in line with prior research in which self-report measures of BA do not correlate with other measurement types such as behavioral measures [45,89]. To date, it has been very difficult to find a sensitive and valid behavioral measure of BA. While it assesses different interoceptive dimensions than the mAMT, the most widely used behavioral measure of interoceptive accuracy is the heartbeat counting task, which has been greatly criticized for relying on non-interoceptive processes [90]. There is a need for valid indirect and behavioral measures of BA and a clear explanation of what aspect of BA they are measuring.

Overall, these findings suggest that BA questionnaires and the mAMT may be measuring both common and different aspects of BA. A refinement of the mAMT is needed, and more research assessing the exact processes at play in the task.

### 4.3. What Is the Impact of an MBI on Transdiagnostic Psychological Processes?

In line with our hypotheses and prior research (see Introduction), MBCT had a significant positive effect on most of the psychological processes measured in the present study; mainly, experiential avoidance (including behavioral avoidance, procrastination, and distraction), unconstructive rumination, emotion regulation self-efficacy and general self-efficacy, actual-ideal self-discrepancy gap and distress, and actual-ought self-discrepancy gap distress in comparison to the control group. Furthermore, as hypothesized, there was no effect of MBCT on the actual-ought self-discrepancy gap. Lastly, contrary to our hypotheses, there was no effect on repression and only a tendential effect on constructive rumination.

Overall, these results support the notion that MBCT is a transdiagnostic intervention that impacts processes common to many psychological disorders [91,92]. For example, experiential avoidance is an important maintenance factor in disorders such as obsessive-compulsive disorder, panic disorder with agoraphobia, borderline personality disorder, depression, and substance use [93]. Likewise, unconstructive rumination has been positively associated with depression, anxiety, co-morbid obsessive-compulsive disorder and generalized anxiety disorder, and borderline personality disorder [94]. With regards to self-efficacy, it has been shown to play an important role in specific phobias and traumatic experiences [70,95]. Last but not least, perceived discrepancies are involved in depression, anxiety, and eating disorders [96]. The influence of MBCT on the aforementioned psychological and transdiagnostic processes may explain its positive effects on a wide array of conditions and populations [97,98]. Whether these processes explain the impact of MBCT on symptomatology is the question we address next.

### 4.4. What Is the Impact of an MBI on Symptomatology, and Is This Effect Mediated by BA and Related Psychological Processes?

As hypothesized, MBCT significantly reduced symptomatology, and this effect was mediated by the regulatory and belief-related aspects of BA (i.e., Attention Regulation, Self-Regulation, Body Listening, and Trusting). This is in line with De Jong and colleagues’ [24] findings showing that BA, as measured by the MAIA, explained the effects of MBCT on depressive symptoms. Unlike the present study, however, De Jong and colleagues [24] found a mediating effect of the facet Not-Distracting only, which may be attributed to their small sample and their specific population (i.e., patients with chronic pain and comorbid active depression). The present study’s findings provide evidence for the long-standing hypothesis that BA is an essential component by which MBCT, and MBIs in general, exert their effects [7]. Here, the aspects of BA that are highlighted as most important are: regulating attention to the body, using body sensations to regulate oneself, and listening to and trusting the body. Therefore, it is not just paying attention to the body that is important but paying attention with an attitude of trusting and using this trusting awareness of the body to regulate one’s distress.

Contrary to our hypotheses, none of the FBSQ facets (i.e., Perception, Differentiation, and Regulation) were significant mediators of the MBCT’s effects on symptomatology. This may be attributed to several reasons. First, the FBSQ is a scale that has not been formally validated and may have some psychometric limitations, meaning that results need to be interpreted with caution. Second, as discussed in the Methods section, the FBSQ and the MAIA evaluate slightly different aspects of BA: the FBSQ focuses on perception, differentiation, and regulation of body sensations during emotional experiences, whereas the MAIA adopts a wider approach and measures attention, attitudes, and reactions towards emotion and non-emotion-related body sensations. Using both measures allowed us to hone in on the nuanced aspects of BA that play a key role in MBIs. The results visibly reveal that it is the attitudinal, regulatory, and attentional aspects across emotional and non-emotional bodily states that explain the effects of an MBI on symptomatology.

Furthermore, multiple mediator models showed that BA (more specifically, the dimensions of Attention Regulation, Self-Regulation, and Body Listening) does not exert its effects alone, but via six other mediators: reduction in experiential avoidance, reduction in unconstructive rumination, enhancement of emotion regulation self-efficacy, reduction of the actual-ideal self-discrepancy gap, and reduction of the distress caused by the actual-ideal and actual-ought self-discrepancy gaps. This suggests that MBCT trains participants to allocate attention to their body with a spirit of listening and regulating, providing the necessary defusion (i.e., a distanced, open, and objective stance towards inner experience) [99] from experience to allow for more functional processes to take place (e.g., adaptive emotion regulation), and to reduce the use of dysfunctional strategies (e.g., experiential avoidance, unconstructive rumination, discrepancy thinking). Altogether, the synergy of these processes explains the effects of MBCT on reduced symptomatology in a heterogeneous clinical sample. These findings are in line with Fissler and colleagues [26], who found that the effect of MBCT on depressive symptoms was explained by an increase in the regulatory and belief-related aspects of BA (as measured by the MAIA) and an increase in the ability to decenter. Results were also in line with the theoretical model of MBCT, in which the central skill is to recognize and disengage from unhelpful states of mind (mainly abstract, repetitive, negative discrepancy-based thinking), which can quickly spiral into a depressive episode [23]. Awareness of the body is an important skill used throughout the program that helps participants disengage from automatic ruminative thinking and enter the so-called “being-mode” in which there is a direct engagement with the present-moment experience of the body. The present study’s findings suggest an important role of BA in this process. Finally, our results are in line with recent neurobiological frameworks that suggest that by practicing mindfulness, a transition is observed from the narrative self (activity in the default mode network) to the experiential or embodied self (activity in the salience network) [100,101].

### 4.5. Strengths and Limitations

Three important strengths of the present study are its heterogeneous and representative clinical sample, its multi-modal measurements of the BA construct, and the assessment of a variety of process variables that had not been previously investigated alongside BA. The heterogeneous clinical sample was recruited following routine clinical practice and ensured the study’s ecological validity, thereby contributing to the scarce literature on the effectiveness of MBCT (i.e., its application in routine clinical practice) [102]. Furthermore, the sample supports the study’s external validity in comparison to other RCTs investigating MBCT, as it closely reflects a clinical population who would likely participate in MBCT in the real world [103]. Using different measures of BA allowed for a multi-faceted and nuanced exploration of MBCT’s effect on BA. Last but not least, assessment of a diversity of process variables alongside BA helps advance the literature on the transdiagnostic nature of MBCT and the synergy of mechanisms involved in explaining its effects.

The present study has several limitations that need to be considered. First of all, participants were not randomized to their respective groups, which may have increased bias [104]. Since there was no randomization, it is not possible to rule out extra-therapeutic and nonspecific factors (e.g., motivation and expectation of improvement) that may have led to improvements in the participants [105]. Second, the control group consisted of age-, gender-, and education-matched controls, meaning we cannot know for sure to which degree the effects observed are due to the treatment itself or to other variables that were not controlled for, such as nonspecific treatment factors (e.g., placebo effects, novelty effects, or effort justification) [105]. Furthermore, there were significant baseline differences between the MBCT group and the control group, which may have led to an overestimation of the intervention’s effects. For more adequate comparisons, future studies should employ an active control condition with comparable baseline characteristics to the experimental group. Third, there was some attrition bias as various participants did not complete assessments at Time 2 and were excluded from the analyses. These participants were, on average, younger than participants who completed both assessment points. Fourth, the present study did not include follow-up measurements preventing us from drawing conclusions about whether the effects are preserved in the long term. Fifth, the present study used two unvalidated measures, the mAMT and the FBSQ, which may have influenced the validity of the results obtained with these two measures. Finally, the conclusions that can be drawn from the mediation analyses are limited because they indicate whether covariations are consistent or not with a hypothesized causal model, but they do not ascertain causality [106].

## Figures and Tables

**Figure 1 brainsci-12-00285-f001:**
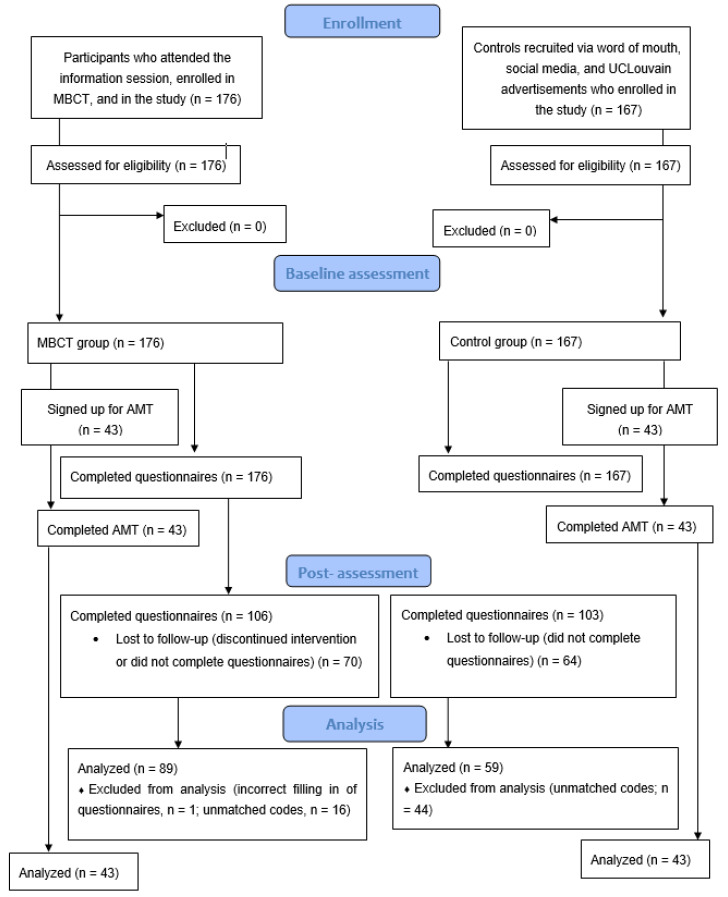
Flow of participants through study.

**Figure 2 brainsci-12-00285-f002:**
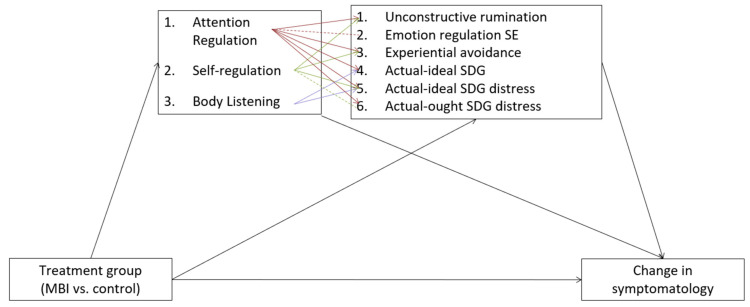
A summary of the multiple mediator models of the effects of treatment on symptomatology through body awareness (as measured by the MAIA) and transdiagnostic psychological processes as described in Section 3.5.2. All variables indicate changes in the variable (pre- and post-change scores). Dashed lines indicate associations that were not robust in the sensitivity analysis. SDG = self-discrepancy gap; SE = self-efficacy.

**Table 1 brainsci-12-00285-t001:** Baseline demographic and clinical characteristics per group.

	MBCT	Control	*t/ᵡ^2^*-Test		
	(*n* = 89)	(*n* = 59)	*t/ᵡ^2^*	*df*	*p*
Age (*M*, *SD*)	43.12 (12.59)	41.03 (16.18)	0.84	103	0.40
Female gender (%)	65.20	71.20	0.34	1	0.56
Education (%)			2.63	2	0.27
Secondary	6.74	13.59			
Higher education (non-university)	24.7	28.80			
Higher education (university)	68.5	57.6			
Employment status (%)			9.02	2	0.01
Employed	66.31	50.83			
Student	6.74	23.7			
Other	26.92	25.4			
Current use of medication (%)	26.98	11.85	3.01	1	0.08

**Table 2 brainsci-12-00285-t002:** Means and standard deviations of variables, stratified by group and measurement occasion.

	MBCT (*n* = 89)		Control (*n* = 59)	
Variable	Baseline *M* (*SD*)	Post *M* (*SD*)	Baseline *M* (*SD*)	Post *M* (*SD*)
**FBSQ Total**	1.41 (0.71) ^a^	2.18 (0.82) ^a^	1.88 (0.87) ^b^	1.92 (0.93) ^a^
Perception	1.41 (0.78) ^a^	2.16 (0.81) ^a^	1.92 (0.91) ^b^	1.91 (0.93) ^a^
Differentiation	1.70 (0.92) ^a^	2.31 (0.94) ^a^	2.14 (0.96) ^b^	2.19 (1.04) ^a^
Regulation	1.11 (0.81) ^a^	2.09 (0.99) ^a^	1.58 (0.95) ^b^	1.64 (0.98) ^b^
**MAIA**				
Noticing	2.44 (1.07) ^a^	3.39 (0.72) ^a^	2.99 (1.26) ^b^	2.91 (1.12) ^b^
Not-Distracting	2.20 (0.98) ^a^	2.64 (0.95) ^a^	2.26 (0.98) ^a^	2.36 (0.97) ^a^
Not-Worrying	2.30 (0.99) ^a^	2.78 (0.83) ^a^	2.34 (1.08) ^a^	2.35 (1.08) ^b^
Attention Regulation	1.76 (0.99) ^a^	3.20 (0.60) ^a^	2.09 (1.06) ^a^	2.11 (1.08) ^b^
Emotional Awareness	2.73 (1.01) ^a^	3.59 (0.80) ^a^	2.94 (1.10) ^a^	2.85 (1.20) ^b^
Self-Regulation	1.63 (0.95) ^a^	3.32 (0.69) ^a^	2.17 (1.08) ^b^	1.93 (1.06) ^b^
Body Listening	1.26 (0.97) ^a^	2.73 (0.81) ^a^	1.84 (1.18) ^b^	1.88 (1.16) ^b^
Trusting	2.15 (1.22) ^a^	3.28 (0.98) ^a^	2.81 (1.21) ^b^	2.89 (1.10) ^b^
**MEAQ Total**	125.76 (27.17) ^a^	103.56 (21.86) ^a^	115.71 (25.67) ^b^	115.75 (25.80) ^b^
Behavioral avoidance	40.89 (10.57) ^a^	32.08 (8.32) ^a^	38.20 (11.11) ^a^	38.49 (12.17) ^b^
Procrastination	24.78 (8.32) ^a^	20.72 (7.32) ^a^	20.59 (7.50) ^b^	19.81 (7.21) ^a^
Dist./Suppression	24.54 (7.36) ^a^	22.13 (6.26) ^a^	24.84 (8.10) ^a^	26.58 (7.63) ^b^
Repression & Denial	35.56 (12.70) ^a^	28.64 (9.31) ^a^	32.09 (10.62) ^a^	30.88 (10.74) ^a^
**Mini-CERTS**				
Unconstructive R	21.00 (4.18) ^a^	17.48 (3.70) ^a^	18.98(3.99) ^b^	18.81 (3.95) ^b^
Constructive R	16.15 (3.16) ^a^	17.48 (2.39) ^a^	16.96 (3.46) ^a^	16.98 (4.10) ^a^
**SE-Q total**	58.11 (12.96) ^a^	72.03 (9.75) ^a^	69.20 (13.43) ^b^	71.70 (12.12) ^a^
Emotion Regulation SE	36.14 (18.31) ^a^	65.37 (14.69) ^a^	54.19 (23.95) ^b^	56.59 (19.22) ^b^
**SD-S**				
Actual-Ideal Gap	4.75 (1.39) ^a^	3.74 (1.23) ^a^	3.77 (1.53) ^b^	3.96 (1.35) ^a^
Actual-Ideal Distress	4.42 (1.57) ^a^	3.55 (1.53) ^a^	2.86 (1.69) ^b^	2.91 (1.81) ^b^
Actual-Ought Gap	4.26 (1.53) ^a^	3.73 (1.25) ^a^	3.23 (1.32) ^b^	3.30 (1.31) ^a^
Actual-Ought Distress	4.15 (1.74) ^a^	3.23 (1.38) ^a^	2.55 (1.56) ^b^	2.58 (1.60) ^b^
**SCL-90-R**	2.00 (0.40) ^a^	1.57 (0.34) ^a^	1.78 (0.57) ^b^	1.73 (0.52) ^b^
**mAMT**				
Specific positive mem	3.44 (1.35) ^a^	4.29 (0.89) ^a^	3.58 (1.05) ^a^	3.56 (1.01) ^b^
Specific negative mem	3.00 (1.40) ^a^	3.69 (1.26) ^a^	2.93 (1.24) ^a^	3.60 (1.24) ^a^
Mention of BS positive	1.21 (1.81) ^a^	2.10 (2.73) ^a^	0.93 (1.30) ^a^	0.86 (1.06) ^b^
Mention of BS negative	1.88 (2.37) ^a^	3.33 (3.27) ^a^	1.42 (1.83) ^a^	1.53 (1.92) ^b^

Note. This table reflects between-group *t*-test comparisons of baseline and post-assessment scores. Per row, variables sharing the same superscript (i.e., ^a^ and ^a^) do not differ at *p* < 0.05. Variables with different superscripts (i.e., ^a^ and ^b^) significantly differ at *p* < 0.05. Mem = memories; mention of BS positive = mention of body sensations during positive memory recall; mention of BS negative = mention of body sensations during negative memory recall.

**Table 3 brainsci-12-00285-t003:** Intervention effect on psychological parameters assessed by random-intercept multilevel models.

Post-Treatment Outcome Measures	Intercept	β(*SE*)—Intervention	β(*SE*)—Baseline Levels
FBSQ Total	0.03 (0.09)	0.61 (0.12) ** *p* = 0.002	0.63 (0.07) *** *p* < 0.001
FBSQ Perception	−0.003 (0.10)	0.57 (0.13) ** *p* = 0.004	0.50 (0.07) *** *p* < 0.001
FBSQ Differentiation	0.04 (0.10)	0.45 (0.13) * *p* = 0.01	0.64 (0.07) *** *p* < 0.001
FBSQ Regulation	0.05 (0.11)	0.71 (0.14) ** *p* = 0.002	0.55 (0.08) *** *p* < 0.001
MAIA Noticing	−0.002 (0.10)	0.67 (0.13) ** *p* = 0.003	0.39 (0.06) *** *p* < 0.001
MAIA Not-Distracting	0.07 (0.12)	0.22 (0.15) *p* = 0.204	0.32 (0.08) *** *p* < 0.001
MAIA Not-Worrying	−0.02 (0.10)	0.38 (0.13) * *p* = 0.025	0.51 (0.06) *** *p <* 0.001
MAIA Attention Regulation	0.06 (0.09)	0.96 (0.13) *** *p* < 0.001	0.41 (0.06) *** *p <* 0.001
MAIA Emotional Awareness	−0.02 (0.10)	0.68 (0.14) ** *p* = 0.003	0.54 (0.06) *** *p* < 0.001
MAIA Self-Regulation	−0.12 (0.10)	1.29 (0.14) *** *p* < 0.001	0.39 (0.06) *** *p* < 0.001
MAIA Body Listening	0.06 (0.11)	0.99 (0.15) *** *p* < 0.001	0.43 (0.07) *** *p* < 0.001
MAIA Trusting	0.12 (0.11)	0.71 (0.14) ** *p* = 0.003	0.47 (0.06) *** *p* < 0.001
MEAQ Total	−0.07 (2.59)	−16.94 (3.39) ** *p* = 0.002	0.53 (0.06) *** *p* < 0.001
MEAQ Behavioral Avoidance	0.20 (1.08)	−7.13 (1.41) ** *p* = 0.002	0.55 (0.06) *** *p* < 0.001
MEAQ Procrastination	−0.80 (0.72)	−2.50 (0.93) * *p* = 0.04	0.62 (0.06) *** *p* < 0.001
MEAQ Dist. /Suppression	1.38 (0.80)	−3.02 (1.04) * *p* = 0.027	0.42 (0.07) *** *p* < 0.001
MEAQ Repression & Denial	−1.02 (2.04)	−4.54 (2.35) *p* = 0.101	0.59 (0.05) *** *p* < 0.001
Mini-CERTS Unconstructive R	−0.12 (0.41)	−2.65 (0.53) ** *p* = 0.003	0.56 (0.06) *** *p* < 0.001
Mini-CERTS Constructive R	0.04 (0.34)	1.05 (0.44) *p* = 0.053	*b* = 0.60 (0.07) *** *p* < 0.001
SCL-90-R total	−0.03 (0.04)	−0.32 (0.06) ** *p* = 0.001	0.58 (0.06) *** *p* < 0.001
SE-Q Total	1.87 (1.14)	8.40 (1.50) ** *p* = 0.001	0.49 (0.06) *** *p* < 0.001
SE-Q Emotion Regulation SE	0.99 (2.00)	17.99 (2.71) *** *p* < 0.001	0.31 (0.06) *** *p* < 0.001
SD-S Actual-Ideal Gap	0.18 (0.16)	−0.87 (0.21) ** *p* = 0.006	0.35 (0.07) *** *p* < 0.001
SD-S Actual-Ideal Gap Distress	0.06 (0.20)	−0.71 (0.26) * *p* = 0.037	0.43 (0.08) *** *p* < 0.001
SD-S Actual-Ought Gap	0.06 (0.15)	−0.47 (0.20) *p* = 0.057	*b* = 0.42 (0.07) *** *p* < 0.001
SD-S Actual-Ought Gap Distress	−0.01 (0.18)	−0.60 (0.24) * *p* = 0.047	0.27 (0.07) *** *p* < 0.001
mAMT’s specific positive memories (with outliers)	−0.02 (0.14)	0.56 (0.20) * *p* = 0.046	0.26 (0.08) ** *p* = 0.002
mAMT’s specific positive memories (without outliers)	−0.02 (0.14)	0.54 (0.20) *p* = 0.06	0.26 (0.08) ** *p* =.003
mAMT specific negative memories	0.49 (0.16)	0.06 (0.23) *p* = 0.808	0.47 (0.09) *** *p* < 0.001
mAMT mention of BS positive	−0.04 (0.30)	0.60 (0.43) *p* = 0.232	0.32 (0.14) * *p* = 0.02
mAMT mention of BS negative	0.07 (0.38)	0.99 (0.54) *p* = 0.14	0.43 (0.13) ** *p* = 0.001

Note. * *p* < 0.05, ** *p* < 0.01, *** *p* < 0.001; R = rumination; SE = self-efficacy; mention of BS positive = mention of body sensations during positive memory recall; mention of BS negative = mention of body sensations during negative memory recall.

## Data Availability

The data presented in this study are available on request from the corresponding author.

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
