# Peer review of "Mindfulness-Based Interventions and Body Awareness"

_brainsci, 2022, doi:10.3390/brainsci12020285_

Round 1

Reviewer 1 Report

Thank you for allowing me to review this interesting study.

1) The authors report a relatively large non-randomized trial of an 8-week MBI investigating the role of bodily awareness in its mechanism of action. The participants in the MBI arm were volunteers interested in MBCT, who had participated in a free information session offered by the Specialized Psychological Consultation Center. This is described (line 691) as a “heterogenous clinical sample”. The baseline global severity of their psychopathological symptoms was 2.00 ±0.40 [scale range 0-4], which was significantly higher than the 1.78 ±0.57 score in the age-, gender-, and education-matched control group. The control group was recruited via social media and advertisement at the university’s psychology faculty. The authors acknowledge (lines 425-6) the significant differences between the 2 groups. The Discussion needs a few sentences whether or not these differences may have influenced the study findings. If the authors collected and can report more specific data about the participant characteristics it would help clarify whether the control group should be characterized as part of the heterogenous clinical sample or as healthy volunteers. As this is a non-randomized trial with groups significantly different at baseline in clinical variables this needs to be a bit clearer.
2) Is it assumed that the participants’ ethnicity is all white?
3) The major drop-off from n =343 baseline participants to 149 8 weeks later needs some explanation and discussion. In a randomized trial this would be included in the details of a CONSORT diagram. Maybe the authors could provide a comparable diagram.
4) I enjoyed the thorough literature review in the introduction and the rationale for the selection of psychological parameters and questionnaires. I am not convinced about lumping the AMT task together with the Heartrate Detection Task as “behavioral measures”. Objective interoceptive accuracy is quite different from reporting body sensations in the recall of personal emotional events. That there is a lack of valid and reliable behavioral/objective measures for interoceptive dimensions is generally agreed upon. As many readers (like me) may not be familiar with the AMT, it would help if it would be more clearly explained how exactly the scores in Tables 2 and 3 are derived.
5) Paragraph 3.3.3 may benefit from including the direction of significant effects.
6) The legend for Table 3 is unclear. Maybe: “the effect of the MBI intervention on psychological parameters assessed by random-intercept multi-level models.”
7) The authors did not report any finding on the instructor level of the models, or did I miss that? That has been quite a topic in recent years and lead to studies assessing the variance between instructors’ teaching qualities in MBI studies.
8) Regarding the six sets of mediation models in section 3.5.2, although models are exploratory and not causally explanatory, I would suggest a table and a graphic depiction of the models. I am aware that the editor may complain about the extra space needed, but it may help the clarity of the reported findings. It is very cumbersome to read through the data-rich paragraphs.
9) I enjoyed reading the discussion. I would extend the limitation section according to the points mentioned above.

Reviewer 2 Report

This study explores the effects of a mindfulness-based intervention (MBCT) on behavioral and self-report measures of body awareness, and the mediating effects of body awareness on symptom severity. Notable strengths of this study are its use of multi-method assessment measures, and detailed attention to mediation effects. Understanding the mechanisms of MBIs is an important contribution to the literature, as it can lead to improvements in interventions and a more fundamental understanding of symptom change. Pre-registration of the hypotheses and analysis plan is also a strength. Limitations include the non-randomized design, sample attrition, and the use of a behavioral task that has not been previously validated. Differential recruitment of the control and intervention group is a significant limitation. The authors appropriately present rationale for the study, clearly describe study methods, and discuss study conclusions in the context of existing literature. I have a few suggestions to help strengthen the manuscript for publication.
1) Introduction
The introduction is thoughtful and well-written. It provides a clear and appropriate overview of the literature relevant to this study. Hypotheses are clearly laid out and well supported.
Lines 116 and 137: Rather than stating, “it would be interesting,” consider use of stronger language to provide rationale for why exploring mechanisms of MBIs and using mixed measures of BA can advance science.
2)Methods
I especially appreciate the use of MLM to address subjects nested in instructors.  The manuscript would be improved with a brief analysis of variance attributable to differences in instructors, as the impact of differential therapist quality/variance in instructors on outcome is of scientific and clinical interest. The intention would not be to significantly lengthen or extend the number of analyses presented, which is already quite extensive.
Line 194: Please whether everyone recruited was a patient at the Specialized Psychological Consultation Center (I assume not), and any information regarding participant rates of mental health diagnoses.
I am not convinced that the use of the control sample as described offers a valid comparision with the MBCT group.  These groups were not selected from the same population, and as demonstrated in Table 2, there are significant baseline differences between the groups in many relevant variables despite demographic matching.  I would be less concerned if it was a nonrandomized waitlist control (same sample, with or without intervention).  It is an apples-to-oranges comparison to say that a group of individuals seeking treatment changed following an intervention, while a group of people who were not seeking treatment didn’t change after not receiving an intervention.  The most compelling results are based on within-subject changes from the intervention group.
3)Results
This could be addressed in either Methods or Results: it would be helpful to have more clarity about how the subset who completed the AMT was selected (presumably it is a convenience sample of participants who were able to schedule within the specified time window following recruitment, but it would be good to make that clear.) Assuming it was convenience selection, it would be helpful to have some indication of any differences between the subsample and the larger sample/potential selection bias.
I am confused by the results as presented in Table 2.  It appears that it is reporting comparisons between baseline data (subscripts a versus b) but not at post (either pre-post within-subjects comparisons, or between-groups comparisons of post scores.) That should be made explicit in the Note for the table, or ideally, the additional comparisons should be reflected in the table. 
4)Conclusion
I believe it is important for the authors to provide hypotheses regarding the lack of correlation between the two self-report measures of BA. Was this expected? I would also be helpful to discuss why only MAIA sub-scales were significant in mediation analyses and none of the FBSQ ones were.
The authors may wish to note as a strength that, although the study is unblinded and unrandomized, it has higher external validity than other RCTs of MBCT given that the sample in this study more closely reflects the population of patients who would be likely to participate in MBCT in the real world. See “The Next Generation of Mindfulness-Based Intervention Research: What have we learned and where are we headed?” by Rosenkranz, Dunne and Davidson for reference.  This strength would be present even if the analysis changed to an open-trial design. 
Line 783: I am curious if the authors have specific recommendations for how the AMT task may be improved
Line: 861 It would be helpful if the authors could expand on how non-randomization could have contributed to bias – e.g. participant motivations and expectations of the effects of MBCT. I also imagine the participants who were most motivated/interested were the ones to sign up for the AMT.
Line 865: The number of participants lost to follow-up is quite significant (~57%). I would request that the authors provide  further discussion of reasons for attrition and how this may have affected study findings.

Round 2

Reviewer 2 Report

Concerns are adequately addressed.

Author Response

Thank you.